# Incorporating Climate Risk into Credit Risk Modeling: An Application in Housing Finance

**Alexandra Lefevre [1,†] and Agnes Tourin [2,\*,†]**

1   Department of Mathematics for Engineering, Courant Institute of Mathematical Sciences, New York University Tandon School of Engineering, 2 MetroTech Center, Room 865, Brooklyn, NY 11201, USA; al6257@nyu.edu
2   Department of Finance and Risk Engineering, New York University Tandon School of Engineering, 6 MetroTech Center, Brooklyn, NY 11201, USA
\*   Correspondence: at1744@nyu.edu
†   These authors contributed equally to this work.

**Abstract:** This paper examines the integration of climate risks into structural credit risk models. We focus on applications in housing finance and argue that mortgage defaults due to climate disasters have different statistical features than default due to household-specific reasons. We propose two models incorporating climate risk based on two separate default definitions. The first focuses on default as a response to a decrease in home value, and the second defines default as a consequence of missed mortgage payments. Using mortgage performance data during Hurricane Harvey, we conduct an empirical study whose results suggest that climate events are potentially another source of undiversifiable credit risk affecting homeowners' ability to make contractual monthly payments. We also show that incorporating this climate-specific default process may capture additional uncertainty in default probability assessments.

**Keywords:** climate risk; credit risk; mortgage default; Merton model; compound Poisson process





## 1. Introduction

With an increasing number of catastrophic events, climate change is becoming impossible to ignore and impacts every economic dimension [1]. This work proposes a systematic approach for incorporating climate risk into credit risk models, and tests these models to conclude whether devastating climate events are a separate source of undiversifiable risk (i.e., cannot be hedged against) using publicly available U.S. conventional mortgage performance data.

Climate risk is defined as financial risk posed by impacts of climate change and how we adapt to these impacts. The former is known as physical risk, referring to the direct damage to infrastructure posed by climate events, and can be further broken down into chronic and acute physical risk. Chronic physical risks are climate changes that occur over a longer period of time, such as rising temperatures and sea levels. Acute physical risk typically refers to sudden and unexpected natural disasters such as hurricanes and wildfires. The latter is known as transition risk, typically referring to business-related risks that follow societal and economic shifts toward a low-carbon and more climate-friendly future, including policy and regulatory risks, technological risks, market risks, reputational risks, and legal risks. Climate risk also affects a borrower's repayment ability, i.e., credit risk.

The U.S. housing market faces many of the same climate risks posed in other industries [2]. Physical risk is the most difficult to ignore, with visible effects on housing property along coastlines, as well as from other extreme climate events such as wildfires. The housing market is also unique in that overall risk is distributed and affects many investors. In the U.S. housing market, about 70% of mortgages are "conventional", i.e., meet underwriting

standards such that they can be pooled and securitized into Mortgage-Backed Securities (MBS). Therefore, mortgage default directly affects returns on MBS. Mortgage default risk is shouldered by homeowners, lenders, servicers, and Government-Sponsored Enterprises (GSEs), and can be mitigated through guarantee fees, loan loss reserves, and risk-based pricing [2]. If a default occurs, GSEs are required to cover losses in securitized pools; they will try to recover through collateral, but physical damage to the property and neighborhood will limit such recoveries. This is especially true if many properties are affected at once (as is usually the case with a natural disaster). Guarantee fees help mitigate climate-related default risk; however, it is difficult to adjust these fees, as insurance requirements quickly become outdated, with affected areas rapidly evolving [3]. Effective loan loss reserves are dependent on successfully forecasting expected future losses due to climate and agreeing on methodologies across regulators, accounting firms, and financial institutions. Risk-based pricing approaches might be the most promising, but currently these do not incorporate climate events. Credit risk models determine an individual borrower's likelihood of making mortgage payments and are used in underwriting procedures. The characteristics used by lenders to evaluate credit risk aim to include equity-related factors that predict the probability of a default, including variables such as loan-to-value ratio, expected home value appreciation, the age of the loan, and the interest rate [4]. Climate events are very difficult to accurately price into these models since insurance companies rely on the calibration on historic climate events, along with scenario projections.

Furthermore, natural disasters can affect both property values (housing price risk) and mortgage payments made (climate credit risk). The latter has a much better documented relationship with default, with income losses or inability to restore a home due to property and infrastructure damage, or not having adequate insurance [5]; however, the former cannot be ignored in the context of climate events. Therefore, the goal of this study is to investigate climate impacts on credit risk defined in terms of property values (wealth) and mortgage payments (cashflows), and how the default process changes across these formulations.

Most work on integrating climate risk focuses on macroeconomic impacts, mainly through scenario studies and simulations [2,6]. One particular challenge is estimating damages and discerning between climate and non-climate events, as well as calibrating the impact size without any historical data. Uncertainty tends to be highly fat-tailed for aggregate welfare impacts of climate change [7]. Furthermore, the private sector has been leading the initiative to incorporate climate risk into credit risk assessments to evaluate individual borrowers and mitigate losses [8]. However, these approaches tend to be sector-specific and proprietary, with no way to capture risk transfer strategies. As of now, financial markets do not systematically and transparently incorporate climate credit risk [8].

Transition and physical risks both impact physical infrastructure, and the US Housing Market is particularly vulnerable to the latter. Refs. [5,9] show that Hurricane Sandy and Hurricane Harvey have been linked to increased borrower delinquency. Refs. [10,11] show that borrowers might not have a full understanding of flood risk. Insurance programs tend to be highly strained and natural disaster scores are sorely outdated, all suggesting that government-supported disaster recovery is not sustainable into the future [2,12]. Practically, default on a mortgage is defined as three missed monthly payments. However, the effect of decreasing home values on the default rate has been studied as well. Ref. [13] finds that negative home equity has a stronger relationship with default in households that are more borrowing-constrained. Climate risk has a well-documented impact on home values. Refs. [14,15] find that extreme climate events such as Hurricane Sandy lead to persistent negative equity impacts on homes damaged by these events. Ref. [16] finds that smaller climate events still depress home values, but these negative impacts do not persist. There is a consensus that climate risk is not accurately priced into insurance premiums. Furthermore, there has been previous work [17] showing that climate events lead to a change in securitization dynamics in the housing market, where lenders are more

likely to securitize mortgages in the aftermath of a natural disaster, thereby transferring credit risk. Recently, [18,19] found that costs of climate change might be mis-priced in the U.S. mortgage market and several publications investigated how climate risk is priced into various U.S. equities—[20–23] all concluded that climate-related risk might not be effectively priced into U.S. stocks.

In this work, we focus on climate events' impacts on house prices and repayment ability separately. To build our credit risk models, we look to a rich body of work on credit risk modeling. Standard credit risk models typically fall under structural or reduced form formulations [24]. For the purposes of this paper, structural models are preferable as they capture economic mechanisms for default. Merton [25] assumed idiosyncratic jumps in individual prices, and assumed that 'jump risk' was diversifiable via premiums. Others have extended Merton's work with different distributional assumptions [26–28]; however, we find that Merton's framework is sufficient given our data and can be used with any sector, loan type, and climate risk definition.

In this work, we separate out the "loan" (cashflows) and the "valu" (housing value) aspects, as climate events have a relationship with both in an undetermined way. Specifically, we consider two simple continuous-time quantitative models that were first introduced recently in [29]. The first model focuses on default as a response to a decrease in home value, and is essentially the same as the standard Merton structural credit risk model [25]; we just add a second jump process to the dynamics followed by the home value with the aim of capturing shocks due to climate disasters, and differentiate them from jumps that are due to other causes. The second model aims at describing directly the random evolution of the mortgage unpaid balance rather than the home value and defines default as a consequence of three missed mortgage payments. The stochastic terms in this model are also two jump processes, the role of the latter being to encapsulate the impact of climate events. Following [30], we leverage the characteristic exponent to compute the probability of default of our proposed model for home prices by calculating the first four moments, whereas we directly compute these moments for cashflow models. Next, we link our models to mortgage performance data during Hurricane Harvey and conduct an empirical study and hypothesis testing to study their plausibility. The results show that incorporating the climate-specific jump process may capture additional uncertainty in default probability forecasts, and suggest that climate events are potentially another source of undiversifiable credit risk affecting homeowners' ability to make contractual monthly payments.

The rest of the article is organized as follows. Section 2 covers model formulations, mortgage data and methodology. Section 2.1 is dedicated to credit risk models for housing values, Section 2.2 to credit risk models for mortgage repayments, and Section 2.3 describes the data and methodology of the empirical study. Then Section 3 presents the results. Finally, Section 4 discusses the findings and limitations of the study and concludes.

## 2. Materials and Methods

In this section, we propose simple mathematical models, and describe the mortgage data and the methodology of the empirical study. First of all, we present two groups of mortgage default models based on different definitions of a default event, the former based on home values in Section 2.1, and the latter based on cashflows in Section 2.2. Next, Section 2.3 contains the description of the mortgage data and the methodology.

### 2.1. Climate Effect on Housing Wealth

We use the Merton jump diffusion model [25] to describe movements in house prices in Section 2.1.1; the drift and diffusion components capture, respectively, the deterministic trend of an asset's price process and the noise around that trend, while the jump component captures unexpected movements. In the case of mortgages, these jumps in home prices can occur due to new constructions in the area, sudden changes in crime rate or tax rate, as well as climate events. To separate out the latter, in Section 2.1.2, we extend the Merton Jump Diffusion model to include an additional jump process. We use a separate jump process

because natural disasters tend to be a sudden shock that affects many households at once, a different dynamic to other more typical events affecting default, such as job loss. In the following two sections, we describe the house price and default processes, and derive the default probability as well as the first four standardized moments for these two model specifications.

### 2.1.1. Jump Diffusion Model

We start with the credit risk model [31], which is used as a building block in many structural credit risk models. Denoting by $(\Omega, \mathcal{F}, \mathbb{P})$ the probability space and by $V(t)$, the home value, at time $t \in [0, T]$, where $T > 0$ is a given time horizon, we suppose that the evolution of $V$ is modeled as a Geometric Brownian Motion with constant coefficients, i.e.,

$$\frac{dV(t)}{V(t)} = \mu dt + \sigma dW(t), \tag{1}$$

where $W(t)$ is a Wiener process under $\mathbb{P}$, the drift rate $\mu$ is constant, and the constant $\sigma > 0$ denotes the volatility coefficient.

The home value $V(t)$ can be written in closed form as

$$\ln V(t) = \ln V(0) + (\mu - \frac{1}{2}\sigma^2)t + \sigma W(t).$$

We assume that default occurs when the log-returns on housing value $V(t)$ falls below a predetermined threshold $\Theta$, i.e., when there exists a time $t \in [0, T]$, such that

$$D(t) = \ln \frac{V(t)}{V(0)} = (\mu - \frac{1}{2}\sigma^2)t + \sigma W(t) < \Theta. \tag{2}$$

Furthermore, the default process is normally distributed, i.e., $D(t) \sim N((\mu - \frac{1}{2}\sigma^2)t, \sigma^2 t)$. The probability of default, which describes the likelihood that a borrower will fail to pay back a debt at given time $t$, is written analytically as

$$\mathbb{P}(D(t)) < \Theta) \quad = \quad \Phi\left(\frac{\Theta - (\mu - \frac{1}{2}\sigma^2)t}{\sigma\sqrt{t}}\right), \tag{3}$$

where $\Phi$ denotes the standard normal cumulative distribution function. Since the distribution of the default process is normal, its skewness and excess kurtosis are both 0.

Next, we add a Compound Poisson process to the above model in order to capture abrupt price changes, and encode them in the credit risk model as undiversifiable risk, i.e., price changes due to climate events cannot be hedged against and must be covered via insurance premiums. And so we extend the model to the jump-diffusion approach as described in [25]:

$$\frac{dV(t)}{V(t)} \quad = \quad (\mu - \lambda\nu)dt + \sigma dW(t) + dY(t), \tag{4}$$

where $Y$ is a compound Poisson process, i.e.,

$$Y(t) = \sum_{i=1}^{N(t)} \Pi_i,$$

where $N$ denotes the Poisson process with rate $\lambda > 0$, and $\Pi_i$ denotes the random size of the Compound Poisson process's $i$th jump. We also assume that the jump sizes $(\Pi_i)_i$ are independent and identically distributed with common mean $\nu = \mathbb{E}[\Pi_i]$, and that $\Pi_i$ is independent of $N(t)$, for every positive integer $i$, and every $t \in (0, T]$.

We specify further the distribution of the jump sizes, by setting

$$\ln{(\Pi_i + 1)} \sim N(\mu_\Pi, \sigma_\Pi^2).$$

Note that the parameters must satisfy the relation $\mathbb{E}(\Pi_i) = \nu = e^{\mu_\Pi + \frac{\sigma_\Pi^2}{2}} - 1$.

In the above model, the compound Poisson process does not distinguish whether jumps are due to non-climate or climate events. Writing the home log price as

$$\ln V(t) = \ln V(0) + (\mu - \lambda\nu - \frac{1}{2}\sigma^2)t + \sigma W(t) + \sum_{i=1}^{N(t)} \ln{(\Pi_i + 1)},$$

we define the default process $D(t) = ln\frac{V(t)}{V(0)}$, and write it in closed form as

$$D(t) = (\mu - \lambda\nu - \frac{1}{2}\sigma^2)t + \sigma W(t) + \sum_{i=1}^{N(t)} \ln{(\Pi_i + 1)}. \tag{5}$$

$D(t)$ is non-normal due to the included jump process. Since $\ln{(\Pi_i + 1)} \sim N(\mu_\Pi, \sigma_\Pi)$, conditioning on $N(t) = i$ jumps, it follows that

$$D(t)|N(t) = i \sim N((\mu - \frac{\sigma^2}{2} - \lambda\nu)t + i\mu_\Pi, \sigma^2 t + i\sigma_\Pi^2).$$

The probability of default in a jump diffusion process can be written as the following converging series:

$$\mathbb{P}(D(t) < \Theta) = \quad \sum_{i=0}^{\infty} \frac{exp(-\lambda t)(\lambda t)^i}{i!} \cdot \Phi\left(\frac{\Theta - (\mu - \frac{\sigma^2}{2} - \lambda\nu)t - i\mu_\Pi}{\sqrt{\sigma^2 t + i\sigma_\Pi^2}}\right). \tag{6}$$

The characteristic exponent of the process $D(t)$ is given in [30].

$$\psi(\omega) = i\omega(\mu - \frac{\sigma^2}{2} - \lambda\nu) - \frac{\sigma^2\omega^2}{2} + \lambda e^{i\omega\mu_\Pi - \frac{\sigma_\Pi^2\omega^2}{2}} - 1.$$

It leads to the first four moments for the Merton Jump Diffusion Model for home values

$$m_1(t) = \mathbb{E}(D(t)) = \kappa_1, m_2(t) = Var(D(t)) = \kappa_2,$$
$$m_3(t) = Skewness(D(t)) = \frac{\kappa_3}{\kappa_2^{3/2}}, m_4(t) = Kurtosis(D(t)) = \frac{\kappa_4}{\kappa_2^2},$$

where

$$\kappa_1 = t(\mu - \frac{\sigma^2}{2} - \lambda\nu + \lambda\mu_\Pi),$$
$$\kappa_2 = t(\sigma^2 + \lambda(\sigma_\Pi^2 + \mu_\Pi^2)),$$
$$\kappa_3 = t\lambda(3\mu_\Pi\sigma_\Pi^2 + \mu_\Pi^3),$$
$$\kappa_4 = t\lambda(3\sigma_\Pi^4 + 6\mu_\Pi^2\sigma_\Pi^2 + \mu_\Pi^4).$$

In contrast to the first model, the distribution of the default process at time $t$ is skewed and exhibits excess kurtosis.

### 2.1.2. Climate Risk in Jump Diffusion Model

In an attempt to separate climate-related defaults from defaults that are due to other causes, we incorporate a second compound Poisson process in the home value process.

After a natural disaster, home prices might fall in a given community at the same time, giving rise to a different jump process than the one already included. Integrating climate disaster-related jumps into (5) leads to a third model for the home value $V(t)$, at time $t$.

$$\frac{dV(t)}{V(t)} = (\mu - \lambda v - \alpha v)dt + \sigma dW(t) + dY(t) + dC(t), \qquad (7)$$

where $\mu, \sigma, \lambda, v, Y$, and $W$ have been defined earlier, and $C$ is the compound Poisson process

$$C(t) = \sum_{j=1}^{M(t)} \Gamma_j,$$

where $M$ denotes a Poisson process with rate $\alpha > 0$ which is defined on the same probability space as $N$ relative to the same filtration and is independent of $N$, and $\Gamma_j$ denotes the size of the compound Poisson process's $j$th jump.

Furthermore, we assume that the variables $\Gamma_j$ are independent and identically distributed, with common mean $\mathbb{E}[\Gamma_j] = v$, and that $\Gamma_j$ is independent of $N(t), M(t)$ and of $\Pi_i$, for all $t \in (0, T]$ and all positive integers $i, j$. In the above model, $C(t)$ is meant to capture climate-related jumps in home value whereas $Y(t)$ should capture non-climate-related jumps. Furthermore, we also assume to simplify that $\Gamma_j$ is log-normally distributed with $\ln(\Gamma_j + 1) \sim N(\mu_\Gamma, \sigma_\Gamma^2)$, and as for the other compound Poisson process, we have the relation

$$v = \mathbb{E}(\Gamma_j) = e^{\mu_\Gamma + \frac{\sigma_\Gamma^2}{2}} - 1.$$

The third model leads to the default process

$$D(t) = (\mu - \lambda v - \alpha v - \frac{1}{2}\sigma^2)t + \sigma W(t) + \sum_{i=1}^{N(t)} \ln(\Pi_i + 1) + \sum_{j=1}^{M(t)} \ln(\Gamma_j + 1). \qquad (8)$$

Again, since $\ln(\Pi_i + 1) \sim N(\mu_\Pi, \sigma_\Pi)$ and $\ln(\Gamma_j + 1) \sim N(\mu_\Gamma, \sigma_\Gamma)$, it follows that $D(t)|N(t) = i, M(t) = j \sim N((\mu - \frac{1}{2}\sigma^2 - \lambda v - \alpha v)t + i\mu_\Pi + j\mu_\Gamma, \sigma^2 t + i\sigma_\Pi^2 + j\sigma_\Gamma^2)$. The probability of default is then given by

$$\mathbb{P}(D(t) < \Theta) = \sum_{i=0}^{\infty} \sum_{j=0}^{\infty} \left(\frac{exp(-\lambda t)(\lambda t)^i}{i!}\right) \left(\frac{exp(-\alpha t)(\alpha t)^j}{j!}\right) \qquad (9)$$

$$\cdot \Phi\left(\frac{\Theta - (\mu - \frac{\sigma^2}{2} - \lambda v - \alpha v)t - i\mu_\Pi - j\mu_\Gamma}{\sqrt{\sigma^2 t + i\sigma_\Pi^2 + j\sigma_\Gamma^2}}\right). \qquad (10)$$

The characteristic exponent of $D(t)$ is (see [30])

$$\psi(\omega) = i\omega(\mu - \lambda v - \alpha v - \frac{1}{2}\sigma^2) - \frac{\sigma^2 \omega^2}{2} +$$

$$(\lambda + \alpha)(\frac{\lambda}{\lambda + \alpha}e^{i\mu_\Pi \omega - \frac{\sigma_\Pi^2 \omega^2}{2}} + \frac{\alpha}{\lambda + \alpha}e^{i\mu_\Gamma \omega - \frac{\sigma_\Gamma^2 \omega^2}{2}} - 1).$$

The first four moments of the default process $D(t)$ in the third model are given by

$$m_1(t) = \mathbb{E}(D(t)) = \kappa_1, m_2(t) = Var(D(t)) = \kappa_2,$$

$$m_3(t) = Skewness(D(t)) = \frac{\kappa_3}{\kappa_2^{3/2}}, m_4(t) = Kurtosis(D(t)) = \frac{\kappa_4}{\kappa_2^2},$$

where

$$\kappa_1 = t(\mu - \frac{1}{2}\sigma^2 - \lambda v - \alpha v + \lambda \mu_\Pi + \alpha \mu_\Gamma),$$
$$\kappa_2 = t(\sigma^2 + \lambda(\mu_\Pi^2 + \sigma_\Pi^2) + \alpha(\mu_\Gamma^2 + \sigma_\Gamma^2)),$$
$$\kappa_3 = t(\lambda(\mu_\Pi^3 + 3\mu_\Pi\sigma_\Pi^2) + \alpha(\mu_\Gamma^3 + 3\mu_\Gamma\sigma_\Gamma^2)),$$
$$\kappa_4 = t(\lambda(\mu_\Pi^4 + 6\mu_\Pi^2\sigma_\Pi^2 + 3\sigma_\Pi^4) + \alpha(\mu_\Gamma^4 + 6\mu_\Gamma^2\sigma_\Gamma^2 + 3\sigma_\Gamma^4)).$$

## 2.2. Climate Risk Effect on Cashflows

While the previous section focuses on default with respect to a decrease in underlying asset value, this section proposes a credit risk model based on repayments on a underlying asset; this default definition is particularly important as it follows the definition used by lenders to determine when to initiate the foreclosure process.

### 2.2.1. Secured Loan Model with Sudden and Unexpected Events

We consider a home mortgage with term $T > 0$ and we denote by $L^d(t)$ the mortgage outstanding balance at time $t$. As in [32], if we ignore the risk of default, we can define $L^d(t)$ in continuous time as

$$L^d(t) = L(0)\frac{1 - e^{-r(T-t)}}{1 - e^{-rT}}, \tag{11}$$

where $L(0)$ is the amount owed at initiation, also known as the loan's principal, and $r$ is the monthly interest rate obtained by dividing the annualized rate by 12. Denoting by $p$ the constant monthly payment amount, we have the relation

$$p = \frac{rL(0)}{(1 - e^{-rT})}.$$

We begin by deriving the Ordinary Differential Equation for (11), yielding

$$dL^d(t) = rL^d(t)dt - \frac{rL(0)}{1 - e^{-rT}}dt.$$

In the above equation, the first term represents the continuously compounded interest, whereas the second term captures the continuous flow of constant mortgage payments. Furthermore, we incorporate here the risk of default by adding an arithmetic compound Poisson process $Q$ to the dynamics of the process $L^d(t)$. Similar to the reasoning in the house price model, default typically tends to be driven by household-level factors (e.g., job loss, unexpected medical expenses, etc.), whereas a natural disaster will affect many households' ability to pay all at once, leading to a jump process with differing dynamics. We denote by $(\Omega, \mathcal{F}, \mathbb{P})$ the probability space and by $L(t)$ the resulting outstanding debt process at time $t$, which satisfies the dynamics

$$dL(t) = rL(t)dt - \frac{rL(0)}{1 - e^{-rT}}dt - \lambda v dt + dQ(t), \tag{12}$$

where $Q$ is the compound Poisson process $Q(t) = \sum_{i=1}^{N(t)} Y_i$, $N$ is a Poisson process with rate $\lambda$, $Y_i$ is a random variable representing the size of the $i$th jump of $Q(t)$, and $v$ is the expected jump size, i.e., $v = \mathbb{E}[Y_i] \in \mathbb{R}$. We assume that the variables $Y_i$ are independent and identically distributed, independent of $N(t)$, for all integers $i$ and time $t \in (0, T]$ and that the first four moments of $Y_i$ are finite. The added Compound Poisson term makes the model in integral form similar to the aggregate loss model that is well-known in insurance (see [33] pp. 334–335), and the following calculation will make this fact clearer.

Indeed, the discounted outstanding debt balance satisfies

$$d(e^{-rt}L(t)) = -\frac{rL(0)}{1 - e^{-rT}}e^{-rt}dt - \lambda \nu e^{-rt}dt + e^{-rt}dQ(t),$$

which can be rewritten in integral form as

$$e^{-rt}L(t) = L(0) - \frac{rL(0)}{1 - e^{-rT}}\int_0^t e^{-rs}ds - \lambda \nu \int_0^t e^{-rs}ds + \int_0^t e^{-rs}dQ(s),$$

which finally yields

$$
\begin{aligned}
L(t) &= e^{rt}L(0) + \frac{L(0)(1 - e^{rt})}{1 - e^{-rT}} + \frac{\lambda \nu}{r}(1 - e^{rt}) + e^{rt}\sum_{i=1}^{N(t)} e^{-rS_i}Y_i \\
&= L(0)\frac{1 - e^{-r(T-t)}}{1 - e^{-rT}} + \frac{\lambda \nu}{r}(1 - e^{rt}) + e^{rt}\sum_{i=1}^{N(t)} e^{-rS_i}Y_i \\
&= L^d(t) + \frac{\lambda \nu}{r}(1 - e^{rt}) + e^{rt}\sum_{i=1}^{N(t)} e^{-rS_i}Y_i,
\end{aligned}
\tag{13}
$$

where $S_i$ represents the arrival time of the jump number $i$. From now on, we also assume that $Y_i$ is independent of $S_i$, for all positive integers $i$. Furthermore, we know that $S_i$ has a gamma distribution with parameters $i$ and $\lambda$, i.e., its density function is given by

$$f_{S_i}(t) = \lambda e^{-\lambda t}\frac{(\lambda t)^{i-1}}{(i-1)!}.$$

We consider the default probability

$$\mathbb{P}[L(t) - L^d(t) \geq \Theta p],$$

where $\Theta > 0$ is a given threshold and $p$ is the monthly payment. In the data, $\Theta = 3$—in other words, a mortgage is considered in default once 3 monthly payments have been missed.

And so given (13), we have derived the default process

$$D(t) = \frac{1}{p}(L(t) - L^d(t)) = \frac{1}{p}\left(\frac{\lambda \nu}{r}(1 - e^{rt}) + e^{rt}\sum_{i=1}^{N(t)} e^{-rS_i}Y_i\right).\tag{14}$$

Using the total expectation rule, we can calculate the moments of $D(t)$ (14) (see detailed calculations in the Appendix A):

$$m_1(t) = \mathbb{E}(D(t)) = \kappa_1, m_2(t) = Var(D(t)) = \kappa_2,$$
$$m_3(t) = Skewness(D(t)) = \frac{\kappa_3}{\kappa_2^{3/2}}, m_4(t) = Kurtosis(D(t)) = \frac{\kappa_4}{\kappa_2^2},$$

where

$$\kappa_1 = 0,$$
$$\kappa_2 = \frac{1}{p^2}\left(\frac{\lambda}{2r}\mathbb{E}[Y_i^2](e^{2rt} - 1)\right),$$
$$\kappa_3 = \frac{1}{p^3}\left(\mathbb{E}[Y_i^3]\frac{\lambda}{3r}(e^{3rt} - 1)\right),$$
$$\kappa_4 = \frac{1}{p^4}\left(\frac{\lambda}{4r}\mathbb{E}[Y_i^4](e^{4rt} - 1)\right).$$

#### 2.2.2. Climate Risk in Secured Loan Model

We incorporate the climate risk by adding another arithmetic compound Poisson process to the dynamics of the process $L^d(t)$ described in (12). We denote the resulting outstanding debt process at time $t$ by $L(t)$ and assume now that it satisfies the dynamics

$$dL(t) \quad = \quad rL(t)dt - \frac{rL(0)}{1 - e^{-rT}}dt - \lambda vdt + dQ(t) - \alpha vdt + dC(t), \qquad (15)$$

where $Q(t)$ is as defined in (14), and $C$ is the compound Poisson process defined as $C(t) = \sum_{j=1}^{M(t)} X_i$, where $M$ is a Poisson process with rate $\alpha$, which is defined on the same probability space as $N$, relative to the same filtration and is independent of $N$, and $X_j$ is a random variable representing the size of the $j$th jump of $C(t)$. We again suppose that the variables $X_j$ are independent and identically distributed with common mean $v = \mathbb{E}[X_j] \in \mathbb{R}$, and also independent of $N(t)$, $M(t)$, and $Y_i$, for all integers $i, j$ and time $t \in (0, T]$.

Similar to the above section, the closed form is given by

$$L(t) = L^d(t) + \frac{\lambda v + \alpha v}{r}(1 - e^{rt}) + e^{rt}(\sum_{i=1}^{N(t)} e^{-rS_i}Y_i + \sum_{j=1}^{M(t)} e^{-rR_j}X_j), \qquad (16)$$

where $R_j$ represents the arrival time of the jump number $j$ of the compound Poisson process $C$. We assume that the random variables $Y_i$ and $X_j$ are independent of $R_j$, for all $i, j \in \mathbb{N}$. Then the default process is defined as

$$D(t) = \frac{1}{p}(\frac{\lambda v + \alpha v}{r}(1 - e^{rt}) + e^{rt}(\sum_{i=1}^{N(t)} e^{-rS_i}Y_i + \sum_{j=1}^{M(t)} e^{-rR_j}X_j)), \qquad (17)$$

and subsequent moments are given below.

The first four moments of the default process $D(t)$ are given by

$$m_1(t) = \mathbb{E}(D(t)) = \kappa_1, m_2(t) = Var(D(t)) = \kappa_2,$$
$$m_3(t) = Skewness(D(t)) = \frac{\kappa_3}{\kappa_2^{3/2}}, m_4(t) = Kurtosis(D(t)) = \frac{\kappa_4}{\kappa_2^2},$$

where

$$\kappa_1 = 0$$
$$\kappa_2 = \frac{1}{p^2}(\frac{\lambda}{2r}\mathbb{E}[Y_i^2](e^{2rt} - 1) + \frac{\alpha}{2r}\mathbb{E}[X_j^2](e^{2rt} - 1))$$
$$\kappa_3 = \frac{1}{p^3}(\mathbb{E}[Y_i^3]\frac{\lambda}{3r}(e^{3rt} - 1) + \mathbb{E}[X_j^3]\frac{\alpha}{3r}(e^{3rt} - 1))$$
$$\kappa_4 = \frac{1}{p^4}(\frac{\lambda}{4r}\mathbb{E}[Y_i^4](e^{4rt} - 1) + \frac{\alpha}{4r}\mathbb{E}[X_j^4](e^{4rt} - 1)).$$

*2.3. Data and Empirical Methodology of the Study*

2.3.1. Description of the Data

We use publicly available Freddie Mac (FHLMC) mortgage origination and performance data on single-family homes to evaluate the above formulations. Freddie Mac is one of two GSEs that securitizes mortgages into guaranteed MBS, and provides loan-level credit performance data on all mortgages they purchased or guaranteed from 1999 to 2021. Their Standard Dataset includes single-family fixed-rate conventional mortgages, typically meeting securitization criteria (see [34] for details on loan types and the role of GSEs in the U.S. Housing Market). Using this dataset, we can observe loan-level origination data and monthly performance data for mortgages that meet the GSEs' securitization standards.

Origination data refer to variables used to undersign a loan (e.g. a borrower's credit score, debt-to-income (DTI), and loan-to-value (LTV) measures), mortgage contract details (e.g. time-to-maturity, interest rate, and loan amount), and property characteristics (e.g. home value and geographic data). Monthly performance data capture information on a mortgage at a given point in time; this typically entails data on the amount of payments made (delinquency), modifications made to the loan, and snapshots of mortgage characteristics such as unpaid balances, remaining time to maturity, and current property value.

We design our study around Hurricane Harvey, a Category 4 hurricane that struck Texas and Louisiana in August 2017. Hundreds of thousands of homes were flooded and approximately 70% of homeowners were uninsured [35], causing approximately USD 42.5 billion in property damage [36]. We consider the universe of loans originated in Texas from 2000 to 2017 with associated performance data from January 2017 to December 2018 in Freddie Mac's Single-Family Standard Dataset. We extract two samples from this universe to evaluate our proposed models: the non-default sample aims to capture the 'ideal' processes, i.e., outstanding loan balance and property home values as the bank intended when extending the loan. The climate default sample includes households with an observed disaster-related missed payment and who also defaulted within 6 months of Harvey, since many homes that did not qualify for disaster delinquency might have still defaulted due to the climate event. A household can apply for a delinquency due to disaster if the borrower experiences a financial hardship impacting his or her ability to pay the contractual monthly amount when (1) the property securing the mortgage loan experienced an insured loss, (2) the property securing the mortgage loan is located in a FEMA-Declared Disaster Area eligible for Individual Assistance, or (3) the borrower's place of employment is located in a FEMA-Declared Disaster Area eligible for Individual Assistance. Defaults due to climate can also occur indirectly and are not captured by the explicit 'Disaster Delinquency' flag in the data; a household may never apply due to delinquency disaster aid because their home was not damaged, but there might be significant damage to the neighborhood infrastructure, commercial properties, etc., that also affects the borrower's ability and willingness to repay. Delays in insurance payouts and outdated flood zone maps may also decrease eligibility for aid; only 10% of flooded structures in counties with FEMA declarations in Texas during Hurricane Harvey had National Flood Insurance Policy (NFIP) insurance [3]. Even those who had insurance during Harvey faced payout delays; more than three months after Harvey hit, nearly half of Houston residents stated that they still were experiencing financial or housing-related challenges, including loss of income or ability to repair their home [37]. And so many households might have defaulted due to climate-related reasons that were not marked as delinquent due to disaster. A total of 2000 households are randomly selected for the non-default and climate default definitions to create sizeable and representative samples while avoiding computational constraints (see Table 1 for sample details).

**Table 1.** Description of sample definitions.

| Sample Name | Description | Number of Households | Number of Records |
|---|---|---|---|
| Non-default sample | loans that have never missed a payment. Zero Balance Code = 1 (Prepaid or Matured (Voluntary Payoff)). | 2000 | 47,296 |
| Climate default sample | loans that have been in default due to a climate ('Disaster Delinquency' = 1 or default within 6 months of Hurricane Harvey) | 2000 | 35,021 |

In order to validate that the second sample we constructed has different statistical features than the loans that have defaulted and were not included in the second sample, we show the default rate trends in Figure 1. We define the default rate as the number of mortgages that missed more than three payments over all active loans. The left figure

shows the default rate in our universe of considered loans from 2000 to 2018, comparable to those reported in [38]. Note the increase from 2% to 3% in the default rate shortly following Hurricane Harvey. The middle figure shows the sharp increase in share of households reporting disaster-related delinquency in August 2017 that remained elevated through 2018. The right-most figure aims to compare the default rate due to Hurricane Harvey (i.e., the climate default sample) to non-Harvey-related default. This figure articulates visually that the categorization of default due to non-climate- versus climate-related events is imperfect, but still captures two different processes.

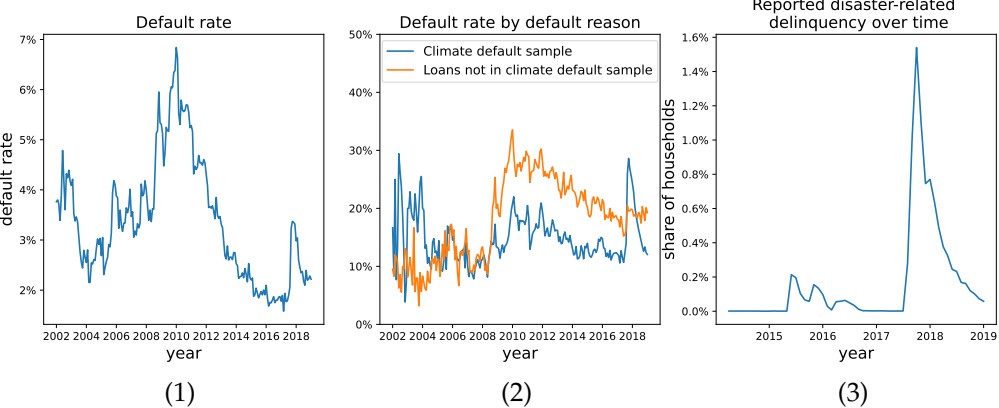

**Figure 1.** From left to right: (**1**) monthly default rates in Texas for loans originated 2000–2017, with monthly performance data 2017–2019, (**2**) comparison of monthly default rates for loans in climate sample versus loans that have defaulted but are not in the climate default sample, and (**3**) share of households reporting climate-related delinquency per month for the same loans as in (**1**).

### 2.3.2. Methodology

We use FHLMC data to evaluate whether or not the models above are specified correctly given the observed data. For both house prices and cashflow models, we explicitly connect the data to our proposed models, estimate parameters, compare specifications, test for the presence of climate-specific effects, and finally compare probability of default curves using the moments derived in the previous sections.

Turning first to the housing price models proposed in Section 2.1, recall that we have defined the default process $D(t) = \ln \frac{V(t)}{V(0)}$. To measure $V(t)$, we use reported property values in the data (refer to Table 2 for full details on how variables in the FHLMC data are used in this analysis). Then, we observe log-returns of home values $\ln \frac{V(t)}{V(0)}$ across households from January 2017 to December 2018 for each sample in Figure 2; $\ln \frac{V(t)}{V(0)}$ seems to follow a bimodal distribution, with the group with decreasing home values much more pronounced for the climate default sample. It is worth adding that these log-returns actually pass the Shapiro–Wilk test for a normal distribution; however, as illustrated in Figure 3, the distribution of the log returns at the single household level is heavy-tailed for most households in each sample.

Next, for the purpose of testing whether there are climate-related jumps in the data, we use the credit risk models introduced in Section 2.1, which are based on default being defined as the log-returns on home value falling below a predetermined threshold. $D(t)$ with parameters $\theta = \{\mu, \sigma, \{\lambda, \mu_\Pi, \sigma_\Pi\}, \{\alpha, \mu_\Gamma, \sigma_\Gamma\}\}$ describes a household-specific default process, so we propose the following approach. Suppose we have $N$ households, and that for each household $n \in 1, ..., N$, we have a separate default process $D^n(t)$, we estimate household-level parameters using Maximum Likelihood Estimation (MLE) and acquire a vector of household-level estimates $\{\theta^n\}$ for each sample. In Section 3, we report and compare the mean estimated coefficients and their associated 95% confidence intervals across samples and model specifications. It is worth acknowledging that the likelihood functions above are not well-behaved, a given when we are expecting discontinuous jumps.

Furthermore, the sample sizes are very small at the household level, with some households having less that 20 observations. Following the MLE estimation technique proposed in [39], we define the following hypotheses to test:

$$H_0^{\text{house price}} : D^n(t) = (\mu - \frac{1}{2}\sigma^2)t + \sigma W(t)$$

$$H_1^{\text{house price}} : D^n(t) = (\mu - \lambda\nu - \frac{1}{2}\sigma^2)t + \sigma W(t) + \sum_{i=1}^{N(t)} \ln(\Pi_i + 1)$$

$$H_2^{\text{house price}} : D^n(t) = (\mu - \lambda\nu - \alpha\nu - \frac{1}{2}\sigma^2)t + \sigma W(t) + \sum_{i=1}^{N(t)} \ln(\Pi_i + 1) + \sum_{j=1}^{M(t)} \ln(\Gamma_j + 1)$$

**Table 2.** Variable field names and calculations. Note that $D(t)$ is used to describe two different default processes, depending on whether a cashflow or a house price model is being considered. The variables.

| Variable | Notation | Calculation in FHLMC Data |
|---|---|---|
| Monthly interest rate (percentage) | $r$ | "Original Interest Rate"/12 |
| Time to Maturity (months) | $T$ | "Original Loan Term" |
| Time since origination (months) | $t$ | "Loan Age" |
| Original Property Value (USD) | $V(0)$ | 'Original Property Value' |
| Actual Property Value (USD) | $V(t)$ | 'Current Actual UPB' / 'Estimated Loan-to-Value (ELTV)' |
| Original Loan Balance (USD) | $L(0)$ | 'Original UPB' |
| Actual outstanding loan balance (USD) | $L(t)$ | 'Current Actual UPB' |
| Expected outstanding loan balance (USD) | $L^d(t)$ | $L^d(t) = L(0)\frac{1-e^{-r(T-t)}}{1-e^{-rT}}$ |
| Monthly payment (USD) | $p$ | $\frac{rL(0)}{1-e^{-rT}}$ |
| Default at time $t$ in house price model | $D(t)$ | $\ln\frac{V_t}{V_0}$ |
| Default at time $t$ in cashflows model | $D(t)$ | $\frac{1}{p}(L(t) - L^d(t))$ |
| Number of climate disaster-related jumps | $M(t)$ | $\#\{\|\frac{L(t)-L^d(t)}{p}\| \geq$ 1\| delinquency during Harvey observed in data$\}$ |
| Number of non-climate disaster-related jumps | $N(t)$ | $\#\{\|\frac{L(t)-L^d(t)}{p}\| \geq 1\|M(t) = M(t-1)\}$ |
| Expected non-climate-related jump size (USD) | $Y_i$ | $p * sign(\frac{L(t)-L^d(t)}{p}\|N(t) > N(t-1)), sign(x) =$ $\begin{cases} -1, & x < 0 \\ 0, & x = 0 \\ 1, & x > 0 \end{cases}$ |
| Expected climate-related jump size (USD) | $X_j$ | $p * sign(\frac{L(t)-L^d(t)}{p}\|M(t) > M(t-1))$ |

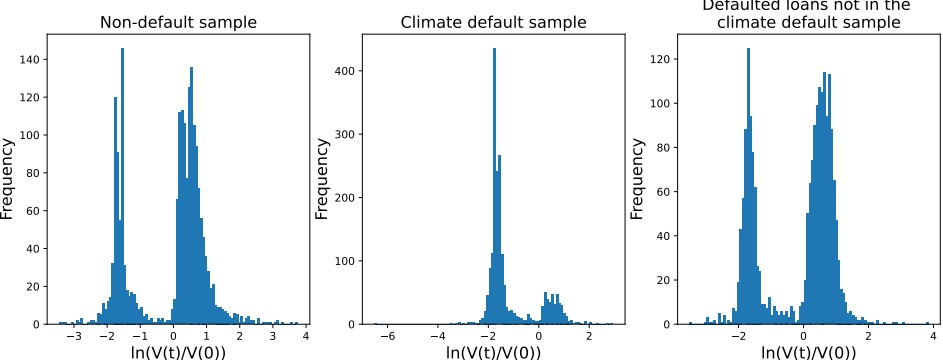

**Figure 2.** Distribution of log-returns across households for each sample 2017–2019 (with log-returns averaged within each household over the time period the household is observed in the data).

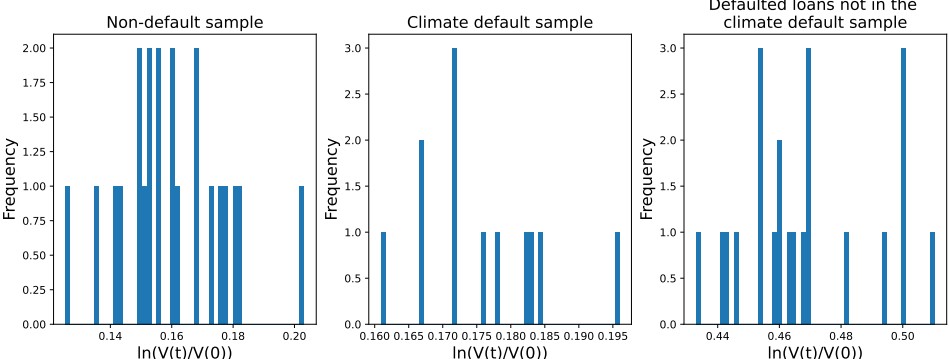

**Figure 3.** Distribution of log-returns for a sampled household for each sample 2017–2019.

Using (3), (6), and (10), we define corresponding (log-)likelihood functions used in the optimization procedure, where $K$ denotes the number of observations in a given household:

$$\text{Log L}_0 = \log \prod_{k=1}^{K} \Phi \left( \frac{D^n(t) - (\mu - \frac{1}{2}\sigma)t}{\sigma\sqrt{t}} \right)$$

$$\text{Log L}_1 = \log \prod_{k=1}^{K} \sum_{i=1}^{\infty} \frac{e^{-\lambda t}(\lambda t)^i}{i!} \Phi \left( \frac{D^n(t) - (\mu - \frac{1}{2}\sigma - \lambda\nu)t - i\mu_\Pi}{\sqrt{\sigma^2 t + i\sigma_\Pi^2}} \right)$$

$$\text{Log L}_2 = \log \prod_{k=1}^{K} \sum_{i=1}^{\infty} \sum_{j=1}^{\infty} \frac{e^{-\lambda t}(\lambda t)^i}{i!} \frac{e^{-\alpha t}(\alpha t)^j}{j!} \Phi \left( \frac{D^n(t) - (\mu - \frac{1}{2}\sigma - \lambda\nu - \alpha\nu)t - i\mu_\Pi - j\mu_\Gamma}{\sqrt{\sigma^2 t + i\sigma_\Pi^2 + j\sigma_\Gamma^2}} \right)$$

In practice, $-2\text{Log L}$ is reported instead of Log L, and so a good fit results in a high Log L and low $-2\text{Log L}$ (with a perfect fit resulting in Log L = 1, $-2\text{Log L} = 0$). The continuous time variable is approximated by discrete intervals of 1-month length. Note that we have truncated the number of jumps in the above likelihood functions. We estimate the upper bound on the number of jumps in the summation to be 3 for non-climate jumps and 1 for climate jumps; we have run experiments for different values of the upper bound (up to 10), with estimations stabilizing with the chosen values. Moreover, this choice is in agreement with our intuition; indeed, homes are not valuated often, and climate jumps have been historically rare (in this case we are considering one single climate event).

Next, we test for the best-fitting sample–model pair using the mean Log L returned under each hypothesis and a Likelihood Ratio (LR) test to evaluates the goodness-of-fit. With nested models, we use the simpler model as the null hypothesis and the model with additional terms as the alternate hypothesis. If the ratio is sufficiently small, we reject the simpler model. In this case, we compare $H_0^{\text{house price}}$, $H_1^{\text{house price}}$, and $H_2^{\text{house price}}$ for each sample using the maximum likelihoods Log $L_0$, Log $L_1$, and Log $L_2$. We conclude our testing for climate jumps with comparing the default probability curves by computing the first four moments as described in earlier sections.

Secondly, we turn to the cashflow models of Section 2.2. For the cashflows models, the default process has been defined earlier as $D(t) = \frac{1}{p}(L(t) - L^d(t))$. The outstanding loan balance that is provided in the monthly data corresponds to the time-discretized variable $L(t)$ with time intervals of 1-month length. Furthermore, the correspondence between the data and $L^d(t)$ is shown in Table 2. Recall that $M(t)$ signifies a Poisson process for climate disaster-related jumps. To estimate $M(t)$, we count the number of times a payment is not equal to the scheduled payment amount, given a household reported delinquency due to a natural disaster. We similarly define $N(t)$, after excluding the jumps used to estimate $M(t)$.

Given that $D(t)$ is a household-level process, we again consider the set of default processes $\{D^n(t)|\theta^n\}$, where parameters $\theta^n = \{\{v, \lambda\}, \{v, \alpha\}\}$. In accordance with (14) and (17), we define our hypotheses for the debt process $D^n(t)$ as

$$H_0^{\text{cashflow}} : D^n(t) = 0$$

$$H_1^{\text{cashflow}} : D^n(t) = \frac{1}{p}\left(\frac{\lambda v}{r}(1 - e^{rt}) + e^{rt}\left(\sum_{i=1}^{N(t)} e^{-rS_i}Y_i\right)\right)$$

$$H_2^{\text{cashflow}} : D^n(t) = \frac{1}{p}\left(\frac{\lambda v + \alpha v}{r}(1 - e^{rt}) + e^{rt}\left(\sum_{i=1}^{N(t)} e^{-rS_i}Y_i + \sum_{j=1}^{M(t)} e^{-rR_j}X_j\right)\right)$$

We assume that there is a maximum of one jump per reporting period (1 month).

On the one hand, a visual inspection of the histograms of the jump sizes $X$ and $Y$ (which represent any deviation from the scheduled payments) across all the households and for each sample (see Figure 4) suggests that these variables have a continuous distribution with a wide range. Consequently, we postulate that $X$ and $Y$ can be considered normally distributed and we validate this hypothesis using the Shapiro–Wilk test. On the other hand, since we did not identify the distribution of $D(t)$, it is natural to consider using the Generalized Method of Moments (GMM), which makes no distributional assumptions and is widely used for economics and finance applications (see [40] for details on GMM). In the end, we chose to design a two-step procedure combining MLE and GMM. First, we use MLE to estimate $v$ (and $v$) by fitting a normal distribution to the jump sizes $Y$ (and $X$), and secondly, we apply a GMM procedure to estimate the remaining parameters $\lambda$ (and $\alpha$). Specifically, the moment conditions $m(\theta^n)$ are functions of the parameters $\theta^n = \{\lambda, \alpha\}$, and GMM finds parameter estimates such that $m(\hat{\theta}^n) = 0$. To test whether the model is correctly specified, we can check whether the estimated parameters $\hat{\theta}^n$ get the moments sufficiently close to zero, i.e., $m(\hat{\theta}^n) \approx 0$. For this purpose, we use the Sargan–Hansen J-test, where the calculated J-statistic is used to test the hypotheses: $h_0 : m(\hat{\theta}^n) = 0$ and $h_a : m(\hat{\theta}^n) \neq 0$, to determine goodness-of-fit of $H_1^{\text{cashflow}}$ and $H_2^{\text{cashflow}}$ with respect to both samples. We also evaluate which of $H_1^{\text{cashflow}}$ and $H_2^{\text{cashflow}}$ is the best model for each sample using the returned J-statistic of an LR test. Note that since $H_0$ has no parameters to estimate and we know that jumps exist in the data (see Table 3), we exclude it from the model comparison. Finally, we compare moments across the three specifications; to compare to $H_0^{\text{cashflow}}$, we calculate $D(t)$ directly from the data and use its sample moments. All analysis was carried out using open-source software in Python and in R, with all code available here: https://bitbucket.org/al6257/credit-risk-modeling/src/master/ (accessed on 3 September 2023).

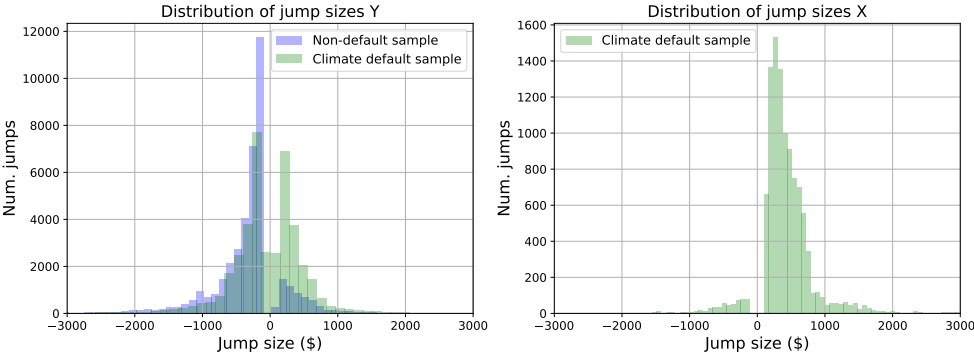

**Figure 4.** Distribution of jump sizes X and Y across all households in each sample.

**Table 3.** Summary statistics of estimated variables from data for cashflow models. Statistics for $D(t), M(t), N(t)$ are taken across households in the last month in the sample (December 2018). Statistics for expected jump sizes $Y$ and $X$ were taken across households one month after Harvey.

| (Variable) | Mean | Stddev | p25 | p50 | p75 |
|---|---|---|---|---|---|
| | | Non-default sample | | | |
| $D(t)$ | −0.254 | 2.41 | −0.126 | −0.0035 | −0.000 |
| $N(t)$ | 11.076 | 18.443 | 1.000 | 3.000 | 13.000 |
| $M(t)$ | 0.00 | 0.00 | 0.00 | 0.00 | 0.00 |
| $Y$ | −305.748 | 1695.460 | −361.619 | −178.072 | 171.080 |
| $X$ | 0.00 | 0.00 | 0.00 | 0.00 | 0.00 |
| | | Climate default sample | | | |
| $D(t)$ | −0.714 | 22.270 | −0.025 | −0.001 | 1.000 |
| $N(t)$ | 16.273 | 22.788 | 1.000 | 7.000 | 21.000 |
| $M(t)$ | 0.300 | 0.772 | 0.00 | 0.00 | 0.00 |
| $Y$ | −567.421 | 8450.797 | −226.714 | 184.968 | 390.500 |
| $X$ | 374.373 | 354.708 | 204.119 | 311.295 | 526.653 |

## 3. Results

First, we compare various mortgage and borrower characteristics across the non-default sample and the climate default sample in Table 4. As expected, the non-default sample has lower DTI and LTV and higher credit score at the average, compared to the climate default sample, all indicators of good financial health. At the average, the non-default sample also has the higher home values $V(0)$ and lower payments $p$ due to lower monthly rates, again highlighting that borrowers in this sample skew towards higher-income and are less liquidity-constrained. Note that the mortgage term T and LTV have small standard deviations and similar values at both the 25th and 75th percentiles because LTV has a strict cutoff when underwriting loans, and most mortgages in the data have a 30-year term. Next, we apply the methodology described in Section 2.3 to conduct hypothesis testing for climate jumps using models specified under Sections 2.1 and 2.2.

**Table 4.** Mortgage and borrower characteristics for each sample, calculated at the beginning of our sample period (January 2017).

| (Variable) | Mean | p25 | p50 | p75 | stddev |
|---|---|---|---|---|---|
| | | Non-default sample | | | |
| DTI | 35 | 27 | 36 | 44 | 9.89 |
| Credit Score | 745 | 715 | 757 | 786 | 42.20 |
| LTV | 72 | 64 | 79 | 81 | 12.909 |
| T | 295 | 180 | 360 | 360 | 83.464 |
| r | 4.09 | 3.62 | 4.00 | 4.50 | 0.65 |
| $V(0)$ | 228,137 | 155,751 | 225,263 | 327,142 | 81,949 |
| p | 997 | 658 | 929 | 1332 | 417 |
| | | Climate default sample | | | |
| DTI | 38 | 33 | 40 | 45 | 7.988 |
| Credit Score | 702 | 671 | 706 | 748 | 43 |
| LTV | 80 | 73 | 80 | 95 | 13.104 |
| T | 320 | 360 | 360 | 360 | 72.197 |
| r | 4.6 | 4.0 | 4.5 | 5.0 | 0.79 |
| $V(0)$ | 222,851 | 152,125 | 222,074 | 318,309 | 81,220 |
| p | 1052 | 694 | 983 | 1425 | 435 |

### 3.1. Housing Wealth Models

We provide a summary of household-level MLE estimates for the housing wealth models in Table 5, where we report the mean estimate across all households along with the 95% confidence interval around the sample mean. Confidence intervals are narrow at the 95th percentile across most parameters, suggesting that estimates do not vary greatly across households. The optimizer returns different estimates for $\{\lambda, \mu_\Pi, \sigma_\Pi\}$ and $\{\alpha, \mu_\Gamma, \sigma_\Gamma\}$ under $H_2^{\text{house price}}$, therefore detecting two independent compound Poisson processes within each sample. The non-default sample has a smaller $\lambda$ and smaller $\mu_\pi$ under $H_1^{\text{house price}}$ than the climate default sample (i.e., both fewer jumps and jumps of smaller magnitude in home value). Also, under $H_2^{\text{house price}}$, the climate default sample has a higher rate of both climate- and non-climate-related jumps $(\lambda, \alpha)$ and larger jump magnitudes $(\mu_\pi, \mu_\Gamma)$ for both Compound Poisson processes, compared to the non-default sample; in addition, the magnitude of the climate-related jumps $(\mu_\Gamma)$ is higher than non-climate-related jumps $(\mu_\pi)$ for both samples. We also observe that Log-likelihood is highest for both samples under $H_2^{\text{house price}}$.

**Table 5.** MLE parameter estimates at the household level. Each estimate is reported as the 95% confidence interval around the sample mean.

| | \multicolumn{6}{c}{*Dependent Variable:*} | | | | | |
| | $H_0^{\text{house price}}$ | | $H_1^{\text{house price}}$ | | $H_2^{\text{house price}}$ | |
| | (Non-Default Sample) | (Climate Default Sample) | (Non-Default Sample) | (Climate Default Sample) | (Non-Default Sample) | (Climate Default Sample) |
|---|---|---|---|---|---|---|
| $\mu$ | $0.088 \pm 0.015$ | $0.03 \pm 0.012$ | $0.535 \pm 0.027$ | $0.822 \pm 0.026$ | $2.873 \pm 0.019$ | $2.778 \pm 0.015$ |
| $\sigma$ | $0.48 \pm 0.022$ | $0.56 \pm 0.018$ | $0.604 \pm 0.022$ | $0.613 \pm 0.018$ | $0.025 \pm 0.005$ | $0.076 \pm 0.013$ |
| $\lambda$ | | | $0.02 \pm 0.002$ | $0.033 \pm 0.003$ | $0.012 \pm 0.001$ | $0.029 \pm 0.003$ |
| $\mu_\pi$ | | | $0.307 \pm 0.022$ | $0.322 \pm 0.019$ | $0.037 \pm 0.005$ | $0.059 \pm 0.006$ |
| $\sigma_\pi$ | | | $0.177 \pm 0.022$ | $0.192 \pm 0.019$ | $0.028 \pm 0.008$ | $0.049 \pm 0.01$ |
| $\alpha$ | | | | | $0.012 \pm 0.001$ | $0.029 \pm 0.003$ |
| $\mu_\Gamma$ | | | | | $0.054 \pm 0.009$ | $0.068 \pm 0.009$ |
| $\sigma_\Gamma$ | | | | | $0.023 \pm 0.005$ | $0.045 \pm 0.007$ |
| $-2 \log L$ | $29.594 \pm 3.536$ | $36.665 \pm 2.859$ | $3.526 \pm 4.112$ | $34.103 \pm 3.01$ | $2.522 \pm 1.402$ | $4.663 \pm 1.229$ |
| $N$ | 1993 | 1738 | 1993 | 1738 | 1993 | 1738 |

Table 6 shows that $H_0^{\text{house price}}$ is the best fit for the non-default sample and $H_2^{\text{house price}}$ is the best fit for the climate default sample. This all points to the fact that there is a separate compound Poisson process present in the data, particularly in the sample with more pronounced climate-related defaults. Table 7 reports the estimated moments of the default curve for each sample, using the mean parameter estimates returned in Table 5. Next, Figure 5 offers a visualization of the distribution of log-returns given our estimated moments. It was produced by using the implementation of the Gram–Charlier expansion of the normal distribution, allowing for the first two moments to coincide with a normal distribution but for higher moments to deviate. In practice, we simulated the log-returns by drawing samples from a log-normal distribution and fitted the expanded pdf with the given moments, allowing for a clean comparison of the curves. $H_0$ seems to estimate log-returns equally for both samples and captures no skewness or excess kurtosis in the data. $H_1^{\text{house price}}$ has the highest kurtosis, thus capturing the most uncertainty for both samples . Excess kurtosis decreases under $H_2^{\text{house price}}$; $H_2^{\text{house price}}$ might absorb some of this uncertainty with the added jump process, accounting for these outliers in a higher second moment. Both $H_1^{\text{house price}}$ and $H_2^{\text{house price}}$ are heavier-tailed than $H_0^{\text{house price}}$.

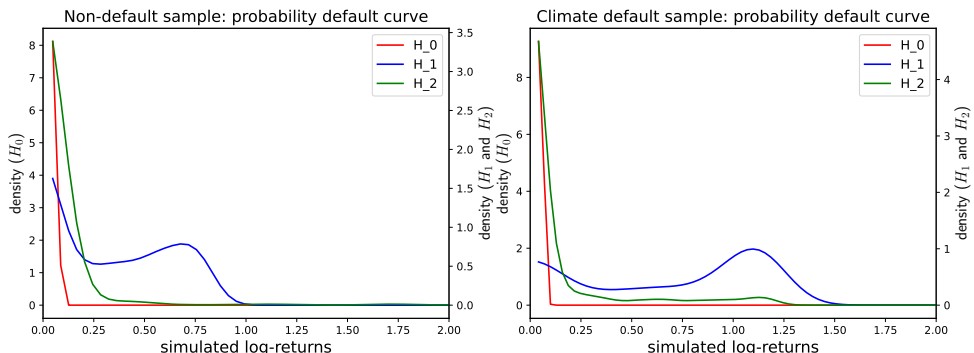

**Figure 5.** Probability curves as given by estimated moments in Table 7. The moments are fitted to simulated random variables following the Gram–Charlier expansion of the normal distribution, in an effort to compare moments and curve shape across samples and specifications.

**Table 6.** (Log-)likelihood ratio test statistics for model comparison, using the mean log-likelihood ratios reported in Table 5.

| | $H_0^{\text{house price}}$ | $H_1^{\text{house price}}$ | $H_2^{\text{house price}}$ |
|---|---|---|---|
| | Non-default sample | | |
| $H_0^{\text{house price}}$ | | 26.068 *** | 27.072 *** |
| $H_1^{\text{house price}}$ | | | 1.004 |
| | Climate default sample | | |
| $H_0^{\text{house price}}$ | | 2.561 | 32.001 *** |
| $H_1^{\text{house price}}$ | | | 29.439 *** |

Note: *** $p < 0.01$.

**Table 7.** Comparison of household-level moments across samples and housing wealth models, reported as the mean moment along with the 95% confidence interval. The mean moments $m_1, m_2, m_3$ and $m_4$ are calculated by taking the temporal average over the entire time frame of the study and the average over all the households of the moments $m_1(t), m_2(t), m_3(t)$ and $m_4(t)$ derived in Sections 2.1.1 and 2.1.2

| | *Dependent Variable:* | | | | | |
|---|---|---|---|---|---|---|
| | $H_0^{\text{house price}}$ | | $H_1^{\text{house price}}$ | | $H_2^{\text{house price}}$ | |
| | (Non-Default Sample) | (Climate Default Sample) | (Non-Default Sample) | (Climate Default Sample) | (Non-Default Sample) | (Climate Default Sample) |
| $m_1$ | $-1.904 \pm 0$ | $-1.904 \pm 0$ | $-0.023 \pm 0.044$ | $0.372 \pm 0.036$ | $2.831 \pm 0.014$ | $2.689 \pm 0.021$ |
| $m_2$ | $1.409 \pm 0$ | $1.409 \pm 0$ | $1.149 \pm 0.061$ | $0.973 \pm 0.03$ | $0.003 \pm 0.001$ | $0.161 \pm 0.02$ |
| $m_3$ | $0 \pm 0$ | $0 \pm 0$ | $0.266 \pm 0.024$ | $0.437 \pm 0.039$ | $0.066 \pm 0.023$ | $0.063 \pm 0.02$ |
| $m_4$ | $0 \pm 0$ | $0 \pm 0$ | $9.921 \pm 0.888$ | $15.073 \pm 1.138$ | $1.992 \pm 0.293$ | $1.446 \pm 0.187$ |

### 3.2. Cashflow Models

We begin by comparing debt levels and jump sizes across the two samples in Table 3. At the average, households have prepaid zero to one monthly payment's worth of outstanding debt ($D(t)$) in both samples. As expected, we observe that $D(t)$ is positive for the climate default sample at higher percentiles, indicating the presence of missed payments. Both samples have negative mean non-climate jumps $Y$, potentially due to prepayments. Even in the climate default samples, households try to repay missed payments to catch up on their loan. This is also reflected in the high number of non-climate-related jumps $N(t)$ in both samples. By contrast, the number of climate-related payment jumps $M(t)$ is much lower

(but higher for the climate default sample than the no-default sample), and the mean climate jump size $X$ is positive, clearly capturing missed payments rather than prepayments.

Table 8 shows a summary of household-level parameter estimates. Confidence intervals are again narrow for most estimates. Comparing the mean parameter estimates, $\nu$ is negative under $H_1^{\text{cashflow}}$ for the climate default sample, potentially capturing more of the prepayment process than a missed payment process. By contrast, under $H_2^{\text{cashflow}}$, it looks like the optimizer estimates the prepayment process in the first jump process (i.e., negative $\nu$) and a missed payment process in the climate-related jump process (i.e., positive $v$). The J-Statistic is very high across both samples under $H_1^{\text{cashflow}}$. However, the statistic drastically decreases under $H_2^{\text{cashflow}}$ for both samples. The statistic is smallest for the climate default sample under $H_2^{\text{cashflow}}$, suggesting that the model under $H_2^{\text{cashflow}}$ for the climate default sample might be the best-specified model, and that there are two different jump processes in the data.

**Table 8.** GMM parameter estimates at the household level. Each estimate is reported as the 95% confidence interval around the mean. The estimates under $H_0$ are calculated as confidence intervals observed directly in the data.

| | $H_0^{\text{cashflow}}$ | | $H_1^{\text{cashflow}}$ | | $H_2^{\text{cashflow}}$ | |
| | (Non-Default Sample) | (Climate Default Sample) | (Non-Default Sample) | (Climate Default Sample) | (Non-Default Sample) | (Climate Default Sample) |
|---|---|---|---|---|---|---|
| $\nu$ | $-1013.855 \pm 28.077$ | $-817.155 \pm 30.511$ | $-252.614 \pm 196.722$ | $-825.915 \pm 252.256$ | $151.617 \pm 40.201$ | $-119.483 \pm 62.178$ |
| $\lambda$ | $0.499 \pm 0.002$ | $0.652 \pm 0.002$ | $0.076 \pm 0.008$ | $0.076 \pm 0.023$ | $0.001 \pm 0$ | $0.061 \pm 0.003$ |
| $v$ | $-0.002 \pm 0.002$ | $30.304 \pm 0.002$ | | | $0 \pm 0$ | $328.343 \pm 0.003$ |
| $\alpha$ | $0 \pm 0$ | $0.037 \pm 0$ | | | $0 \pm 0$ | $0.095 \pm 0.005$ |
| $-2 \log \text{L (Y)}$ | | | $0 \pm 0$ | $58.529 \pm 47.934$ | $0.878 \pm 0.423$ | $23.106 \pm 18.657$ |
| $-2 \log \text{L (X)}$ | | | | | $0 \pm 0$ | $55.379 \pm 1.374$ |
| J-test | | | $46,235,473,254.568 \pm 9,571,620,633.317$ | $12,079,781,867.6 \pm 2,298,005,017.153$ | $0.746 \pm 0.05$ | $0.709 \pm 0.055$ |
| $N$ | 1993 | 1738 | 1993 | 1738 | 1993 | 1738 |

We use Table 9 to compare the models further. Looking at the likelihood ratios, $H_2^{\text{cashflow}}$ is the preferable specification over $H_1^{\text{cashflow}}$ for both samples. Leveraging the coefficients calibrated above, we can compare moments across samples and models as shown in Table 10 and use sample moments to compare $H_0^{\text{cashflow}}$, $H_1^{\text{cashflow}}$, and $H_2^{\text{cashflow}}$. We observe that the second and fourth moments seem inflated under $H_1^{\text{cashflow}}$; because of the large variability of jump sizes (as shown in Table 3), one single Compound Poisson process might not be enough to capture all missed payments and prepayments. This is further supported by the fact that both moments decrease under $H_2^{\text{cashflow}}$, which could indicate that an additional jump process better describes the payment activity in the data, and therefore default. Note that we do not provide a visualization of the moments in contrast to the moments of the house price models in Figure 5, as we did not identify the distribution of $D(t)$.

**Table 9.** Goodness-of-fit analysis across hypotheses for cashflow models.

| | Likelihood Ratio Test J-Statistic ($H_1^{\text{cashflow}}$ versus $H_2^{\text{cashflow}}$) |
|---|---|
| Non-default sample | 46,235,473,254 *** |
| Climate default sample | 12,079,781,867 *** |

Note: *** $p < 0.01$.

**Table 10.** Comparison of household-level moments across samples and cashflow models and reported as the mean moment along with the 95% confidence interval. The mean moments $m_1, m_2, m_3$ and $m_4$ are calculated by taking the temporal average over the entire time frame of the study and the average over all the households of the moments $m_1(t), m_2(t), m_3(t), m_4(t)$ derived in Sections 2.2.1 and 2.2.2

| | $H_0^{\text{cashflow}}$ | | $H_1^{\text{cashflow}}$ | | $H_2^{\text{cashflow}}$ | |
| | (Non-Default Sample) | (Climate Default Sample) | (Non-Default Sample) | (Climate Default Sample) | (Non-Default Sample) | (Climate Default Sample) |
|---|---|---|---|---|---|---|
| $m_1$ | $-9.46 \pm 0.156$ | $-4.088 \pm 0.126$ | $0 \pm 0$ | $0 \pm 0$ | $0 \pm 0$ | $0 \pm 0$ |
| $m_2$ | $1671.687 \pm 37.261$ | $1141.584 \pm 29.673$ | $11,517.493 \pm 411.827$ | $23.456 \pm 0.858$ | $7938.48 \pm 186.212$ | $446.186 \pm 13.442$ |
| $m_3$ | $-3.572 \pm 0.049$ | $-1.427 \pm 0.057$ | $-1832.738 \pm 79.746$ | $5569.317 \pm 178.64$ | $-5.253 \pm 0.059$ | $-0.025 \pm 0.003$ |
| $m_4$ | $20.807 \pm 0.415$ | $13.897 \pm 0.264$ | $10,305,711.142 \pm 514,394.214$ | $91,100,924.95 \pm 3,610,751.915$ | $32.918 \pm 0.671$ | $0.874 \pm 0.027$ |

## 4. Discussion

We proposed two credit risk models incorporating unexpected climate events with default definitions based, respectively, on home values and cashflows. Using mortgage performance data during Hurricane Harvey, we have shown the real-world connection between our proposed models and housing finance data, and were able to successfully compare model specifications, as well as empirically validate our model formulations.

Specifically, we find that the jump diffusion credit risk model, which includes an additional climate default-specific Compound Poisson process in the dynamics followed by the home price, offers a better fit than the original model with a unique jump process. This finding is especially important, as it implies that there is potentially a presence of climate-specific credit risk that is only diversifiable via premiums.

We used MLE for the house-pricing models and a mixture of MLE and GMM for our cashflow models to estimate needed parameters. While statistical discrimination (such as linear regression) and classification methods (supervised learning) are a more standard approach used by lending institutions to assess credit risk for an individual [41], we wanted to use estimation techniques that had minimal assumptions and followed the existing literature on estimating jump risks [39]. A pitfall in our approach is sample size: all households individually have less than 100 observations, and MLE typically requires much larger sample sizes for stable estimates. To address this shortcoming, we have run versions of this analysis with both an increased sample of 5000 households and with diffusion Brownian bridges in an effort to increase the number of observations per household for the house price models without adding any artificial jump (results reported in Appendix B), and our conclusions have remained unchanged. Another potential design limitation is truncating at two Poisson processes; as the goal of this paper is to assess whether or not there are a separate, climate-specific processes, two jumps were sufficient for testing model fit without running the risk of over-fitting.

Even with these limitations, we see that the model specification with two jump processes is the best fit, especially subsetting to data where we know defaults due to climate exist. This model also does a better job in capturing uncertainty in the default process.

We faced similar challenges in evaluating the cashflow model, which also contains a climate default-specific jump process in the dynamics of the unpaid principal balance. Besides the small sample sizes at the household level, another limitation comes from the fact that actual principal payments are not directly observed, and were inferred from the change in unpaid principal balance, the monthly interest rate, maturity of the loan, and the initial unpaid principal balance. Clearly, the presence of reporting errors in the data can affect the accuracy of these calculations, which, in turn, can lead to noisy parameter estimates. However, despite these challenges, the J-statistic is lowest under the model specification with two jump processes for both samples, and at a level indicating that the model is not mis-specified,

suggesting that an additional jump process leads to a better fit. Again, this alludes to the presence of climate-specific credit risk that cannot be hedged against.

This conclusion does not contradict existing literature on integrating climate disasters into financial risk models; [8] finds that integrating climate credit risks into existing frameworks does require changes in lender behavior in terms of which assets to invest in, and [20,23] also find equity risk premiums associated with severe storms and hurricanes, respectively. Our research findings can potentially inform policy discussions. If climate-specific credit risk is undiversifiable, it is better managed through insurance premiums, loan loss reserves, and risk-based pricing. Keeping insurance requirements up-to-date and maintaining adequate loss reserves are two possible policy interventions. Lenders should work on including individual borrower's climate-related default risk in credit risk assessments, in addition to existing determinants of credit risk.

We developed and tested our models within the framework of the U.S. regulatory and banking system; however, our approach is highly flexible and can potentially be applicable to other geographies. This approach can also can be extended to evaluate additional event studies around any type of climate risk, including both chronic physical risk and transition risk, as both are regarded as shock events following a potentially different process, compared to typical jump events affecting assets and loan repayments.

Finally, since defaults due to a natural disaster can happen across households in a short period of time, aggregate estimates across all households should be evaluated in future work and compared to understand market-level dynamics. For example, a single mortgage default might not affect the return on an MBS, but defaults across an entire geographic location around the same time might. Deriving aggregate estimates would allow for an evaluation past single-instrument effects. Future work can also include more rigorous data simulation techniques to generate realistic scenarios, especially if the financial instruments in question do not have high-frequency observations.

**Author Contributions:** Conceptualization, A.L. and A.T.; methodology, A.L. and A.T.; software, A.L.; validation, A.L. and A.T.; formal analysis, A.T.; investigation, A.L.; data curation, A.L.; writing—original draft preparation, A.L. and A.T.; writing—review and editing, A.L. and A.T.; visualization, A.L. All authors have read and agreed to the published version of the manuscript.

**Funding:** This research received no external funding.

**Institutional Review Board Statement:** Not applicable.

**Informed Consent Statement:** Not applicable.

**Data Availability Statement:** Datasets used in this report are publicly available at https://bitbucket.org/al6257/credit-risk-modeling/src/master/samples/ (accessed on 3 September 2023). These samples were built using Freddie Mac Single Family Standard Loan-Level Dataset (documentation, licensing details, and download instructions available here: https://www.freddiemac.com/research/datasets/sf-loanlevel-dataset (accessed on 3 September 2023).

**Conflicts of Interest:** The authors declare no conflict of interest.

**Appendix A**

In this appendix, we compute explicitly the moments of $L(t)$. We start with the fist model that contains only one compound Poisson process. We use the following argument. Conditionally on $N(t) = n$, the variables $S_1, S_2, \cdots, S_n$ are distributed as the ordered values of $n$ independent uniform random variables in $(0, t)$ (see [33], pages 334–335). In other words, $S_1, S_2, \cdots, S_n$ can be interpreted as a random permutation of the random variables $t_1, t_2, \ldots, t_n$ uniformly distributed over the interval $(0, t)$, so that

$$\sum_{i=1}^{N(t)} e^{-rS_n} = \sum_{i=1}^{N(t)} e^{-rt_n}.$$

We can compute the expected outstanding debt by time $t$, $\mathbb{E}[L(t)]$ by using a total expectation rule:

$$\mathbb{E}[L(t)] = L^d(t) + \frac{\lambda\nu}{r}(1 - e^{rt}) + e^{rt}\sum_{k=0}^{\infty}\mathbb{E}[L(t)|N(t) = k)]\mathbb{P}[N(t) = k]$$

$$= L^d(t) + \frac{\lambda\nu}{r}(1 - e^{rt}) + e^{rt}\sum_{k=0}^{\infty}\mathbb{E}[L(t)|N(t) = k)]exp(-\lambda t)\frac{(\lambda t)^k}{k!}$$

$$= L^d(t) + \frac{\lambda\nu}{r}(1 - e^{rt}) + e^{rt}\sum_{k=0}^{\infty}\mathbb{E}[\sum_{i=1}^{N(t)}e^{-rS_i}Y_i|N(t) = k)]exp(-\lambda t)\frac{(\lambda t)^k}{k!}$$

$$= L^d(t) + \frac{\lambda\nu}{r}(1 - e^{rt}) + e^{rt}\sum_{k=0}^{\infty}\mathbb{E}[\sum_{i=1}^{k}e^{-rS_i}Y_i]exp(-\lambda t)\frac{(\lambda t)^k}{k!}$$

$$= L^d(t) + \frac{\lambda\nu}{r}(1 - e^{rt}) + e^{rt}\sum_{k=0}^{\infty}\sum_{i=1}^{k}\mathbb{E}[e^{-rS_i}Y_i]exp(-\lambda t)\frac{(\lambda t)^k}{k!}$$

$$= L^d(t) + \frac{\lambda\nu}{r}(1 - e^{rt}) + e^{rt}\sum_{k=0}^{\infty}\sum_{i=1}^{k}\mathbb{E}[e^{-rS_i}]\mathbb{E}[Y_i]exp(-\lambda t)\frac{(\lambda t)^k}{k!}$$

$$= L^d(t) + \frac{\lambda\nu}{r}(1 - e^{rt}) + e^{rt}\sum_{k=0}^{\infty}\sum_{i=1}^{k}\nu\mathbb{E}[e^{-rS_i}]exp(-\lambda t)\frac{(\lambda t)^k}{k!}$$

$$= L^d(t) + \frac{\lambda\nu}{r}(1 - e^{rt}) + e^{rt}\nu\sum_{k=0}^{\infty}\mathbb{E}[\sum_{i=1}^{k}e^{-rS_i}]exp(-\lambda t)\frac{(\lambda t)^k}{k!}$$

$$= L^d(t) + \frac{\lambda\nu}{r}(1 - e^{rt}) + e^{rt}\nu\sum_{k=0}^{\infty}\mathbb{E}[\Sigma_{i=1}^{k}e^{-rt_i}]exp(-\lambda t)\frac{(\lambda t)^k}{k!}$$

$$= L^d(t) + \frac{\lambda\nu}{r}(1 - e^{rt}) + e^{rt}\nu\sum_{k=0}^{\infty}\sum_{i=1}^{k}\mathbb{E}[e^{-rt_i}]exp(-\lambda t)\frac{(\lambda t)^k}{k!}.$$

Since

$$\mathbb{E}[e^{-rt_i}] = \int_0^t \frac{e^{-rx}}{t}dx = \frac{1}{rt}(1 - e^{-rt}),$$

we finally have

$$\mathbb{E}[L(t)] = L^d(t) + \frac{\lambda\nu}{r}(1 - e^{rt}) + e^{rt}\nu\sum_{k=0}^{\infty}k \cdot \frac{(1 - e^{-rt})}{rt}e^{-\lambda t}\frac{(\lambda t)^k}{k!}$$

$$= L^d(t) + \frac{\lambda\nu}{r}(1 - e^{rt}) + e^{rt}\nu\frac{(1 - e^{-rt})}{rt}\sum_{k=0}^{\infty}k \cdot e^{-\lambda t}\frac{(\lambda t)^k}{k!}$$

$$= L^d(t) + \frac{\lambda\nu}{r}(1 - e^{rt}) + e^{rt}\nu\lambda\frac{(1 - e^{-rt})}{r}$$

$$= L^d(t).$$

Similarly, using the fact that

$$2\sum_{i=1}^{n}\sum_{j=1}^{i-1}e^{-rS_i}e^{-rS_j} = 2\sum_{i=1}^{n}\sum_{j=1}^{i-1}e^{-rt_i}e^{-rt_j},$$

in distribution, we compute

$$
\begin{aligned}
\mathrm{var}[L(t)] &= \mathbb{E}[(L(t) - L^d(t))^2] \\[4pt]
&= \mathbb{E}[(\frac{\lambda\nu}{r}(1 - e^{rt}) + e^{rt}\sum_{i=1}^{N(t)} e^{-rS_i}Y_i)^2] \\[4pt]
&= \sum_{n=0}^{\infty} \mathbb{E}[(\frac{\lambda\nu}{r}(1 - e^{rt}) + e^{rt}\sum_{i=1}^{N(t)} e^{-rS_i}Y_i)^2 | N(t) = n)] \\[4pt]
&\quad \times\ exp(-\lambda t)\frac{(\lambda t)^n}{n!} \\[4pt]
&= \sum_{n=0}^{\infty} \mathbb{E}[(\frac{\lambda\nu}{r}(1 - e^{rt}))^2 + 2(\frac{\lambda\nu}{r}(1 - e^{rt}))e^{rt}\sum_{i=1}^{n} e^{-rS_i}Y_i \\[4pt]
&\quad +\ e^{2rt}(\sum_{i=1}^{n} e^{-rS_i}Y_i)^2]exp(-\lambda t)\frac{(\lambda t)^n}{n!} \\[4pt]
&= (\frac{\lambda\nu}{r}(1 - e^{rt}))^2 + 2(\frac{\lambda\nu}{r}(1 - e^{rt}))e^{rt}\sum_{n=0}^{\infty} \mathbb{E}[\sum_{i=1}^{n} e^{-rS_i}Y_i]exp(-\lambda t)\frac{(\lambda t)^n}{n!} \\[4pt]
&\quad +\ e^{2rt}\sum_{n=0}^{\infty} \mathbb{E}[(\sum_{i=1}^{n} e^{-rS_i}Y_i)^2]exp(-\lambda t)\frac{(\lambda t)^n}{n!} \\[4pt]
&= (\frac{\lambda\nu}{r}(1 - e^{rt}))^2 - 2(\frac{\lambda\nu}{r}(1 - e^{rt}))^2 + e^{2rt}\sum_{n=0}^{\infty} \mathbb{E}[(\sum_{i=1}^{n} e^{-rS_i}Y_i)^2] \\[4pt]
&\quad \times\ exp(-\lambda t)\frac{(\lambda t)^n}{n!} \\[4pt]
&= -(\frac{\lambda\nu}{r}(1 - e^{rt}))^2 + e^{2rt}\sum_{n=0}^{\infty} \mathbb{E}[\sum_{i=1}^{n} (e^{-rS_i}Y_i)^2 \\[4pt]
&\quad +\ 2\sum_{i=1}^{n}\sum_{j=1}^{i-1} e^{-rS_i}e^{-rS_j}Y_iY_j]exp(-\lambda t)\frac{(\lambda t)^n}{n!} \\[4pt]
&= -(\frac{\lambda\nu}{r}(1 - e^{rt}))^2 + e^{2rt}\sum_{n=0}^{\infty} \mathbb{E}[Y_i^2]\sum_{i=1}^{n} \mathbb{E}[e^{-2rS_i}]exp(-\lambda t)\frac{(\lambda t)^n}{n!} \\[4pt]
&\quad +\ 2e^{2rt}\sum_{n=0}^{\infty} \nu^2 \sum_{i=1}^{n}\sum_{j=1}^{i-1} \mathbb{E}[e^{-rS_i}e^{-rS_j}]exp(-\lambda t)\frac{(\lambda t)^n}{n!} \\[4pt]
&= -(\frac{\lambda\nu}{r}(1 - e^{rt}))^2 + (\frac{\lambda\nu}{r}(1 - e^{rt}))^2 + \frac{\lambda}{2r}\mathbb{E}[Y_i^2](e^{2rt} - 1) \\[4pt]
&= \frac{\lambda}{2r}\mathbb{E}[Y_i^2](e^{2rt} - 1).
\end{aligned}
$$

Note that we skipped the arguments that had already been shown in the computation of the expectation of $L(t)$.

Finally, using the fact that, in distribution,

$$
3\sum_{i=1}^{n}\sum_{j=1}^{i-1}(e^{-2rS_i}e^{-rS_j} + e^{-rS_i}e^{-2rS_j}) = 3\sum_{i=1}^{n}\sum_{j=1}^{i-1}(e^{-2rt_i}e^{-rt_j} + e^{-rt_i}e^{-2rt_j}),
$$

and

$$
\sum_{i=1}^{n}\sum_{j=1}^{i-1}\sum_{k=1}^{j-1} e^{-rS_i}e^{-rS_j}e^{-rS_k} = \sum_{i=1}^{n}\sum_{j=1}^{i-1}\sum_{k=1}^{j-1} e^{-rt_i}e^{-rt_j}e^{-rt_k},
$$

we can compute the third moment, which turns out to be

$$
\mathbb{E}[(L(t) - \mathbb{E}[L(t)])^3] = \mathbb{E}[Y_i^3]\frac{\lambda}{3r}(e^{3rt} - 1).
$$

We show below the detailed calculations that use similar techniques as in the calculation of the second moment:

$$
\mathbb{E}[(L(t) - L^d(t))^3]
$$

$$
= \mathbb{E}[(\frac{\lambda v}{r}(1 - e^{rt}) + e^{rt}\sum_{i=1}^{N(t)} e^{-rS_i}Y_i)^3]
$$

$$
= \sum_{n=0}^{\infty} \mathbb{E}[(\frac{\lambda v}{r}(1 - e^{rt}) + e^{rt}\sum_{i=1}^{N(t)} e^{-rS_i}Y_i)^3 | N(t) = n)]exp(-\lambda t)\frac{(\lambda t)^n}{n!}
$$

$$
= \sum_{n=0}^{\infty} \mathbb{E}[(\frac{\lambda v}{r}(1 - e^{rt}))^3 + e^{3rt}(\sum_{i=1}^{n} e^{-rS_i}Y_i)^3
$$

$$
+ \quad 3(\frac{\lambda v}{r}(1 - e^{rt}))^2 e^{rt}(\sum_{i=1}^{n} e^{-rS_i}Y_i) + 3(\frac{\lambda v}{r}(1 - e^{rt}))e^{2rt}(\sum_{i=1}^{n} e^{-rS_i}Y_i)^2]
$$

$$
\times \quad exp(-\lambda t)\frac{(\lambda t)^n}{n!}
$$

$$
= \quad (\frac{\lambda v}{r}(1 - e^{rt}))^3 + \sum_{n=0}^{\infty} \mathbb{E}[e^{3rt}(\sum_{i=1}^{n} e^{-rS_i}Y_i)^3]exp(-\lambda t)\frac{(\lambda t)^n}{n!}
$$

$$
+ \quad -3\frac{\lambda^3 v^3}{r^3}(1 - e^{rt})^3 + 3\frac{\lambda^3 v^3}{r^3}(1 - e^{rt})^3 + \frac{3\lambda^2 v}{2r^2}(1 - e^{rt})\mathbb{E}[Y_i^2](e^{2rt} - 1).
$$

In order to compute the second term, we expand it

$$
\sum_{n=0}^{\infty} \mathbb{E}[e^{3rt}(\sum_{i=1}^{n} e^{-rS_i}Y_i)^3]exp(-\lambda t)\frac{(\lambda t)^n}{n!}
$$

$$
= \quad e^{3rt}\sum_{n=0}^{\infty} exp(-\lambda t)\frac{(\lambda t)^n}{n!}\mathbb{E}[e^{-3rS_i}Y_i^3
$$

$$
+ \quad 3\sum_{i=1}^{n}\sum_{j=1}^{i-1} e^{-2rS_i}Y_i^2 e^{-rS_j}Y_j + 3\sum_{i=1}^{n}\sum_{j=1}^{i-1} e^{-rS_i}Y_i e^{-2rS_j}Y_j^2
$$

$$
+ \quad 6\sum_{i=1}^{n}\sum_{j=1}^{i-1}\sum_{k=1}^{j-1} e^{-rS_i}Y_i e^{-rS_j}Y_j e^{-rS_k}Y_k]
$$

$$
= \quad \mathbb{E}[Y_i^3]\frac{\lambda}{3r}(e^{3rt} - 1) + e^{3rt}\sum_{n=0}^{\infty} exp(-\lambda t)\frac{(\lambda t)^n}{n!}\mathbb{E}[
$$

$$
+ \quad 3\sum_{i=1}^{n}\sum_{j=1}^{i-1} e^{-2rS_i}Y_i^2 e^{-rS_j}Y_j + 3\sum_{i=1}^{n}\sum_{j=1}^{i-1} e^{-rS_i}Y_i e^{-2rS_j}Y_j^2
$$

$$
+ \quad 6\sum_{i=1}^{n}\sum_{j=1}^{i-1}\sum_{k=1}^{j-1} e^{-rS_i}Y_i e^{-rS_j}Y_j e^{-rS_k}Y_k]
$$

$$
= \quad \mathbb{E}[Y_i^3]\frac{\lambda}{3r}(e^{3rt} - 1) + e^{3rt}\sum_{i=0}^{\infty} exp(-\lambda t)\frac{(\lambda t)^n}{n!}
$$

$$
\times \quad \left[3v\mathbb{E}[Y_i^2]\left(\sum_{i=1}^{n}\sum_{j=1}^{i-1} e^{-2rS_i}e^{-rS_j} + \sum_{i=1}^{n}\sum_{j=1}^{i-1} e^{-rS_i}e^{-2rS_j}\right)\right.
$$

$$
+ \quad \left. 6v^3\sum_{i=1}^{n}\sum_{j=1}^{i-1}\sum_{k=1}^{j-1} e^{-rS_i}e^{-rS_j}e^{-rS_k}\right]
$$

$$
= \quad \mathbb{E}[Y_i^3]\frac{\lambda}{3r}(e^{3rt} - 1) + \frac{3v\lambda^2}{2r^2}\mathbb{E}[Y_i^2](e^{2rt} - 1)(e^{rt} - 1) - (\frac{v\lambda}{r}(e^{rt} - 1))^3,
$$

where we used the identities

$$\sum_{l=1}^{N} l = \frac{N(N+1)}{2},$$

and

$$\sum_{l=1}^{N} l^2 = \frac{N(N+1)(2N+1)}{6}$$

to compute the double and triple sums.

Similar arguments and techniques yield the fourth centered moment

$$\mathbb{E}[(L(t) - \mathbb{E}[L(t)])^4] = \frac{\lambda}{4r} \mathbb{E}[Y_i^4](e^{4rt} - 1) + \frac{3}{4} \frac{\lambda^2}{r^2} \mathbb{E}[Y_i^2]^2 (e^{2rt} - 1)^2.$$

In the case of the second model, the arguments are very similar, except for one key difference: We now have two sequences of arrival times in the cross terms relating the two compound Poisson processes, one for the first Poisson process $N$ and the other one for the second Poisson process $M$. However, since the sum of the two Poisson processes is a Poisson process, we can merge these two sequences into one single sequence of arrival times for the sum $N + M$ and compute the moments by using similar arguments as for the first model, in particular, the fact that the sequence of arrival times of the sum of Poisson processes can be viewed as the ordered values of independent uniform random variables. Specifically, given the number of jumps of first type $N(t) = n$ and second type $M(t) = m$, we can use the fact that

$$\sum_{i=1}^{n} \sum_{j=1}^{m} e^{-rS_i} e^{-rR_j} = \frac{1}{2} \Big( \sum_{i=1}^{n+m} \sum_{j=1}^{n+m} e^{-rT_i} e^{-rT_j} - \sum_{i=1}^{n} \sum_{j=1}^{n} e^{-rS_i} e^{-rS_j} - \sum_{i=1}^{m} \sum_{j=1}^{m} e^{-rR_i} e^{-rR_j} \Big),$$

where $T_1, T_2, \cdots, T_j \cdots$ is the sequence of arrival times of $N + M$.

**Appendix B**

We use this appendix to formalize our approach to running the formulations with diffusion bridges for estimating the house price models. To go from monthly to daily observations, we simulate 30 observations each month for each household using a diffusion bridge, with the idea that we can increase the number of observations per household, and within one month the housing value should not change much. To construct the bridge, we use starting value $V_t$ at $t$ and ending value $V_t$ at $t + 1$, with the idea that we preserve the jump at $V_{t+1}$ (i.e., if there was a jump in the original data at $V_{t+1}$, we want to ensure that the algorithm still registers this jump given the increased number of observations immediately before and after the jump). We encode drift as zero and sigma as $c \cdot sqrt(T)$, where $c$ is a small non-zero constant, and $T$ in this case equals 1.

However, as shown in Figures A1 and A2, while the housing value trends are preserved, the MLE algorithm can misinterpret the introduced noise as jumps. This is reflected in the MLE results in Tables A1 and A2, with very high estimated values for $\alpha$ and $\lambda$. Some findings remain consistent with the non-simulated sample. Under $H_2^{\text{house price}}$, the default sample experiences more frequent (larger $\alpha$) and higher-magnitude jumps (larger $\mu_\Gamma$) than the non-default sample. We see an increase in excess kurtosis under $H_2$ for both samples. However, while $H_1^{\text{house price}}$ and $H_2^{\text{house price}}$ are preferred over $H_0^{\text{house price}}$, $H_1^{\text{house price}}$ is preferred over $H_2^{\text{house price}}$ for both samples. This might be due to the increased noise and number of jumps; in Figures A1 and A2, it looks like a single jump process. All in all, this approach led to more stable estimates; however, due to the increase in jumps, the parameter estimates are less interpretable and the model fit deteriorated.

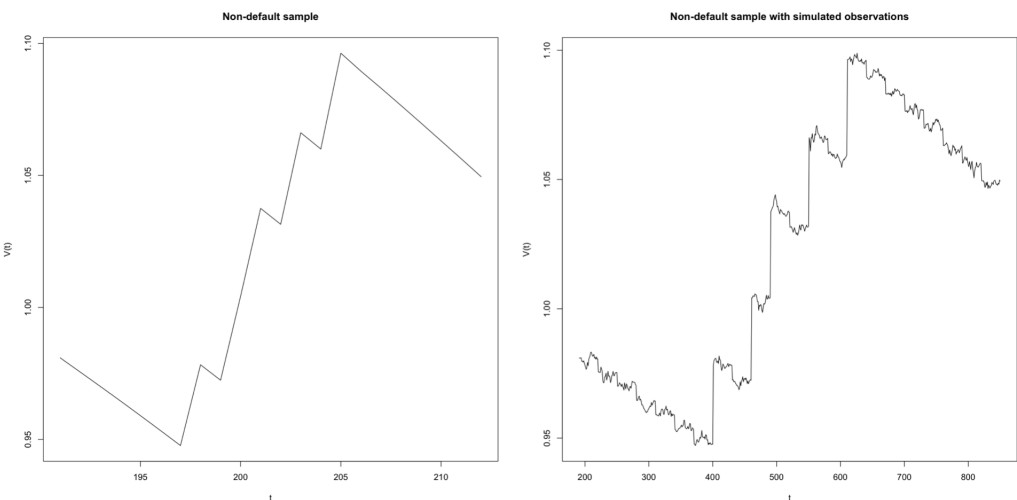

**Figure A1.** Comparison of $V(t)$ in original non-default sample to $V(t)$ simulated with a diffusion bridge.

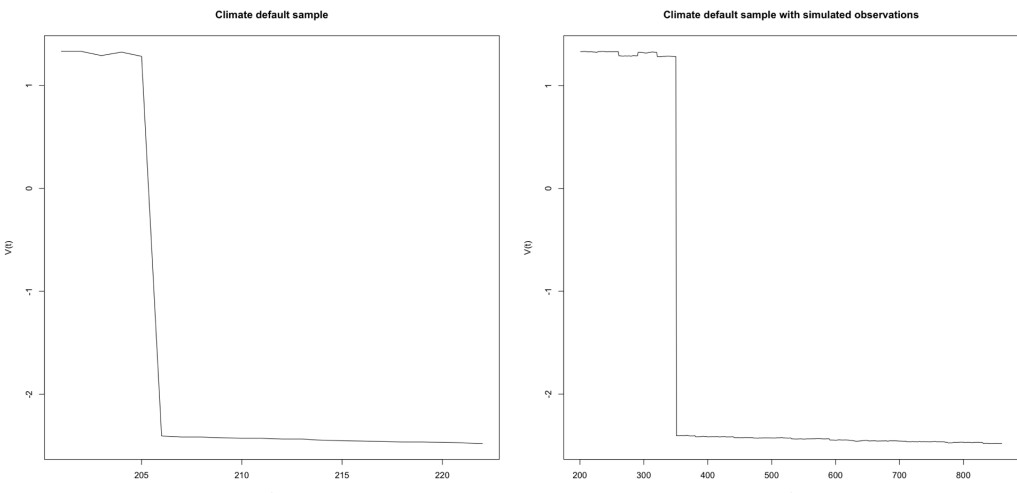

**Figure A2.** Comparison of $V(t)$ in original default sample to $V(t)$ simulated with a diffusion bridge.

**Table A1.** MLE parameter estimates at the household level. Each estimate is reported as the 95% confidence interval around the sample.

| | *Dependent Variable:* | | | | | |
|---|---|---|---|---|---|---|
| | $H_0$ | | $H_1$ | | $H_2$ | |
| | **(Non-Default Sample)** | **(Climate Default Sample)** | **(Non-Default Sample)** | **(Climate Default Sample)** | **(Non-Default Sample)** | |
| $\mu$ | $1.929 \pm 0.027$ | $0.973 \pm 0.027$ | $1.114 \pm 0.001$ | $1.651 \pm 0.41$ | $1.056 \pm 0.001$ | $1.081 \pm 0.013$ |
| $\sigma$ | $1.608 \pm 0.019$ | $1.183 \pm 0.022$ | $1 \pm 0$ | $0.996 \pm 0.002$ | $1 \pm 0$ | $0.997 \pm 0.001$ |
| $\lambda$ | | | $387.699 \pm 1.221$ | $313.653 \pm 2.804$ | $102.745 \pm 0.798$ | $106.285 \pm 0.825$ |
| $\mu_\Pi$ | | | $0.999 \pm 0$ | $1.067 \pm 0.047$ | $1.073 \pm 0.028$ | $1.033 \pm 0.016$ |
| $\sigma_\pi$ | | | $0.999 \pm 0$ | $0.984 \pm 0.002$ | $1.063 \pm 0.027$ | $1.034 \pm 0.016$ |
| $\alpha$ | | | | | $104.474 \pm 0.829$ | $108.538 \pm 0.887$ |
| $\mu_\Gamma$ | | | | | $1.121 \pm 0.038$ | $1.111 \pm 0.038$ |
| $\sigma_\Gamma$ | | | | | $1.126 \pm 0.043$ | $1.084 \pm 0.03$ |
| $-2 \log L$ | $66,757.675 \pm 3373.015$ | $87,579.433 \pm 4024.822$ | $0 \pm 0$ | $175.551 \pm 112.164$ | $274.804 \pm 162.262$ | $147.452 \pm 116.721$ |
| $N$ | 2000 | 2000 | 2000 | 2000 | 2000 | 2000 |

**Table A2.** (Log-)likelihood ratio test statistics for model comparison.

| | $H_0^{\text{house price}}$ | $H_1^{\text{house price}}$ | $H_2^{\text{house price}}$ |
|---|---|---|---|
| | Non-default sample | | |
| $H_0^{\text{house price}}$ | | 4,339,450 *** | 4,339,450 *** |
| $H_1^{\text{house price}}$ | | | 0.000 |
| | Climate default sample | | |
| $H_0^{\text{house price}}$ | | 4,399,457 *** | 4,399,457 *** |
| $H_1^{\text{house price}}$ | | | 0.000 |

Note: *** $p < 0.01$.

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
