# Peer review of "Incorporating Climate Risk into Credit Risk Modeling: An Application in Housing Finance"

_fintech, doi:10.3390/fintech2030034_

Round 1

Reviewer 1 Report

Dear authors,

Thank you very much for your presentation, I really enjoyed reading your article, I think it is scientifically sound, and the underlying philosophy very interesting, I sincerely believe it is also very applicable, however I still have the following comments,

1- Although I see that it works in the US I wonder how the model would react in other geographies?

2- Two different Poisson processes are used (which is really very interesting), but I can't help but wonder if there wouldn't be more underlying Poisson processes, and if the credit risk assessment process shouldn't be split even further ?

3- Given that several drivers have been identified (two here, the traditional scheme and the climat scheme), I wonder how the model would perform versus a regression strategy?

My questions are mere curiosity, as only with the conclusion reached your work deserves to be published.

I hope you'll find these comments valuable.

Best Regards,

Your reviewer

Minor editing (not that relevant)

Reviewer 2 Report

I have several suggestions that the authors should be addressed in the revised manuscript. First) The authors used numerous equations instead of focusing theoretically on the modelling of credit risk. Second) Many studies discussed the determinants of credit risk, which the authors ignored to review it. I recommend adding the recent below studies in this work such as: "The Role of Country Governance in Achieving the Banking Sector’s Sustainability in Vulnerable Environments: New Insight from Emerging Economies. Sustainability, 2023, 15(13), 10538." Third) I suggest separating the conclusion from the discussion section to follow the standard format of scientific articles. Fourth) I recommend explaining the policy implications to the various stakeholders.

It's good.

Reviewer 3 Report

Authors proposed two models/methods of climate risk incorporating to default risk for mortgages/housing. Th3ey proposed two credit risk models incorporating unexpected climate events with default definitions based on first scenario home values and second scenario on cashflows. Using mortgage performance data during Hurricane Harvey, shown the real-world connection between proposed models and housing finance data, and were able to successfully compare model specifications, as well as empirically validate model formulations.

Authors elaborated on theoretical models both types using also Merton model and compound Poisson model. They used not only formulas but also empirical example to proof their concept.

Authors presented and supported their empirical studies with data description, figures and tables.

Tables presented models results. 

All results supported conclusions provided by the end of the article. Discussion concludes this work.

Would apprecaite also adding some more discussion with comapring to other authors/reserachers results from literature. In this form it is more like conclusions but not discussion.

English language is correct and good quality.

Round 2

Reviewer 2 Report

I still suggest expanding the literature review section by explaining the previous studies conducted bout credit risk in particular. This can be reviewed by studies conducted in both emerging and developed markets. This explanation by the authors which the current study performs in the US country does not convince me to ignore credit risk studies in the emerging markets! 41 references is not enough at all.

It's good.